# Clinical Chemistry and Haematology Values of a Captive Population of Humboldt Penguins (*Spheniscus humboldti*)

**DOI:** 10.3390/ani13223570

**Published:** 2023-11-18

**Authors:** Christoph Leineweber, Maike Lücht, Christine Gohl, Hanspeter W. Steinmetz, Rachel E. Marschang

**Affiliations:** 1Laboklin GmbH & Co. KG, Steubenstrasse 4, 97688 Bad Kissingen, Germany; rachel.marschang@gmail.com; 2Tierpark Hellabrunn AG, Tierparkstrasse 30, 81543 Munich, Germany

**Keywords:** Humboldt penguins, *Spheniscus humboldti*, heparin blood, liver values, kidney values, glutamate dehydrogenase, cholinesterase, packed cell volume, white blood cell count

## Abstract

**Simple Summary:**

Blood reference intervals are important for the correct interpretation of blood results but are missing for many wild species. The aim of the present study was therefore to establish specific reference intervals for a series of clinical chemistry and haematology analytes for captive Humboldt penguins (*Spheniscus humboldti*). The results indicate that the factors of sex and age should be considered when interpreting the results of haematological and clinical biochemistry assays.

**Abstract:**

Reference intervals for clinical chemistry and haematology analytes are essential for the interpretation of blood results, but limited data are available for Humboldt penguins (*Spheniscus humboldti*) in zoological collections as well as in the wild. The aim of the present study was therefore to establish reference intervals for a series of clinical chemistry and haematology analytes in a single zoological population of Humboldt penguins. Furthermore, possible variations of the analytes influenced by the age and sex of the individuals were investigated. Lithium heparinized plasma and whole blood samples from 39 animals were analysed and statistically evaluated. The following clinical chemistry analytes were significantly higher in females: glutamate dehydrogenase (*p* = 0.0003), alanine transaminase (*p* = 0.0005), alkaline phosphatase (*p* = 0.0245), aspartate aminotransferase (*p* = 0.0043), creatine kinase (*p* = 0.0016), lactate dehydrogenase (*p* < 0.0001), inorganic phosphorus (*p* = 0.0223), and sodium (*p* = 0.0415). No significant differences between males and females were found for any haematology analyte. Cholesterol (*p* = 0.0233; −0.39), white blood cell count (*p* = 0.0208; −0.40), and absolute heterophiles (*p* = 0.0148; −0.42) had significant negative correlations with the age of the penguins. The results indicate that the factors of sex and age should be considered when interpreting the results of haematological and clinical biochemistry assays, and further studies are needed to determine whether there are more differences in other age groups or seasons.

## 1. Introduction

Humboldt penguins (*Spheniscus humboldti*) are commonly kept in zoological collections around the world, similar to African penguins (*Spheniscus demersus*), but there are only limited data available on blood values of captive Humboldt penguins. In the literature so far, the values published for Humboldt penguins outside ZIMS (Zoological Information Management Software 2023, Species360, 7900 International Drive, Minneapolis, MN 55425, USA) are often from wild animals [1,2] or from animals taken from the wild and kept in a zoo for a short time [3]. The data also consist of a limited number of analytes, especially for clinical chemistry [1,2,3,4]. But, comprehensive reference intervals for clinical chemistry and haematology are important to monitor the health status of captive individuals as well as wild populations. Health assessment is of particular importance since wild populations in Chile and Peru are in decline due to climate change, fishing and harvesting of aquatic resources, industrial pollution, mining and quarrying of minerals and guano, tourism, and habitat loss, as well as diseases (red list of the International Union for Conservation of Nature (IUCN) [5].

There are a variety of clinical chemistry and haematology analytes that are used for the evaluation of the health status of avian patients. The panel should include analytes which are useful to detect major homeostatic abnormalities, metabolic disturbances, and organ damage and dysfunction [6]. Recommended for avian species are electrolytes like total calcium (Ca), inorganic phosphorus (P), sodium (Na), potassium (K), and chloride (Cl); analytes which are associated with major organs like the liver, kidney, pancreas, and muscles; and metabolic analytes like cholesterol, triglycerides, β-hydroxybutyric acid (β-HBS), and uric acid (UA) [6]. Enzymes that have been evaluated for the evaluation of liver cell integrity and the biliary tract include glutamate dehydrogenase (GLDH), gamma-glutamyl transferase (GGT), alanine transaminase (ALT), lactate dehydrogenase (LDH), aspartate aminotransferase (AST), and alkaline phosphatase (AP) [6,7,8]. The creatine kinase (CK) is important for the differentiation between hepatic and muscle injuries because this enzyme is specific for skeletal, cardiac, and smooth muscles and, therefore, serves as a better assessment of the AST and LDH, which are present in muscle and liver cells [6,7,8]. Bile acids (BA) can be used to evaluate liver diseases and liver function [6,7,8]. UA is a good marker for the evaluation of kidney function [6,7]. Amylase can increase in pancreatic disease but can also increase due to enteritis and renal diseases [6,7]. Cholinesterase (CHE) is a biomarker for the detection of organophosphate and carbamate intoxications [9] and can also be used for the evaluation of liver function in birds.

A number of physiological factors can influence blood values in healthy animals. In penguins, studies have documented differences in various blood analytes depending on sex [10,11,12,13], age [12,14], body condition [13], time of day [15], season [13], year [14], location [13], and infection with malaria [16,17].

The goal of the present study was to establish baseline values of healthy captive individuals for a variety of clinical chemistry and haematology analytes for a managed zoological population of Humboldt penguins in Germany at a single time point in winter. It was hypothesized that analytes differ between males and females and vary depending on the age of the individuals.

## 2. Materials and Methods

### 2.1. Animals

The Humboldt penguin colony tested in this study consisted of 43 individuals (22 males, 21 females). Blood was collected from 39 animals (19 males, 20 females) with ages ranging between 3 and 20 years (mean 8 years) and weighing between 2.5 and 5.6 kg (mean 4.31 kg) at the time of blood collection. The penguins were kept in an outdoor enclosure consisting of a terrestrial part of 166.5 m^2^ with concrete and natural stone and a freshwater pool of 148.5 m^2^. The pool was a permanent flow-through well water system, which was drained and cleaned depending on water quality every one to two weeks. Animals were fed a mixture of defrosted sprat (*Sprattus sprattus*), capelin (*Mallotus villosus*), herring (*Clupea harengus*), and mackerels (*Scomber scombrus*) supplemented with vitamin supplements (Mazuri^®^ Fish eater tablets, Mazuri Zoo Foods, Claus GmbH, 67117 Limburgerhof, Germany, 0.5 tablet/animal po three times a week).

All penguins were clinically examined before blood collection and showed no signs of infection or other diseases. The clinical examination included documentation of the body weight and condition, as well as a full external examination of the animals. Exclusion criteria included any abnormalities in the plumage or signs of moderate to severe bumblefoot. Parasitology testing included examination of faecal samples by flotation and evaluation of blood smears for haemoparasites. No parasites were detected in any of the penguins examined.

### 2.2. Sample Collection and Analysis

The blood samples were collected for a clinical health check outside of the breeding (March to June) and moulting (July to August) season in the morning before feeding on the 10th of November 2022. The penguins were separated into smaller groups 15 min before blood collection for faster handling and to avoid stress. Blood samples were collected from the dorsal coccygeal vein under manual restraint within 5 min with 2 mL syringes with a Luer system (HENKE-Ject^®^, Henke-Sass, Wolf GmbH, 78532 Tuttlingen, Germany) and canulae (Terumo^®^ black 22G, 0.7 × 32 mm, Terumo Deutschland GmbH, 65760 Eschborn, Germany) and were transferred to two lithium heparin tubes (Micro sample tube lithium heparin, 1.3 mL, screw cap, SARSTEDT AG & Co. KG, 51588 Nürnbrecht, Germany). Blood smears were prepared and air dried from blood without anticoagulant directly after blood collection. One tube from each animal was centrifuged at 5000× *g* for 5 min (Hettich Mikro 200/200 R, Andreas Hettich GmbH & Co. KG, 78532 Tuttlingen, Germany) no later than two hours after collection, and the plasma was transferred to neutral tubes (Screw cap micro tube, 1.5 mL, SARSTEDT AG & Co. KG, 51588 Nürnbrecht, Germany). The second tube was sent as whole blood for haematology together with the plasma refrigerated (8 °C; 46.4 °F) overnight to the laboratory. The samples were analysed at the time of arrival in the laboratory the next day. No samples with marked to moderate haemolysis or lipemia were included in the analysis. The clinical chemistry analytes amylase, GLDH, GGT, ALT, AP, CHE, BA, AST, CK, LDH, cholesterol, triglycerides, β-HBS, UA, Ca, P, Na, K, and Cl were measured using the cobas^®^ 8000 analyser series module c701 (Roche Diagnostics, 68305 Mannheim, Germany). Lithium heparinized whole blood was used to determine the packed cell volume (PCV): microhaematocrit capillaries were centrifuged for 5 min at 12,000× *g* in a Haemofuge (Thermo Fisher Scientific Inc., 4817 Breda, The Netherlands). Haematology was done using blood smears stained using a Diff-Quick^®^ fast stain kit (Labor+Technik Eberhard Lehmann GmbH, 14167 Berlin, Germany) and evaluated microscopically. Leukocytes were counted at 400× magnification in ten fields of view, and the average of all fields was multiplied by 2.0 [18] to determine the estimated white blood cell count (WBC). Percentual differential counts of heterophils, basophils, eosinophils, lymphocytes, and monocytes were based on evaluation of 100 leukocytes at 1000× magnification using oil emersion [19]. The absolute leucocyte fractions were calculated for each penguin.

### 2.3. Statistical Analyses

The statistical analyses were carried out using the statistical analysis software (SAS 2023, https://welcome.oda.sas.com/, accessed on 10 November 2023) (SAS Institute Inc., Cary, NC, USA). Normal distribution of individual analytes was assessed based on histogram and Q-Q plots, evaluation of whether all values were within two standard deviations, and the Shapiro–Wilk test. The recommendations of the ASVCP guidelines for reference intervals [20] were used for the calculation of reference intervals, and the Reference Value Advisor [21] was used to calculate the lower and upper reference intervals by Box-Cox transformed data. Outliers were determined using the Tukey and the Dixon–Reed tests. The mean, median, SD, minimum, and maximum were also calculated. The statistical effects of sex were calculated by the *t*-test for normally distributed parameters and by the Wilcoxon test for non-normally distributed parameters. Age-specific correlations of the analytes were tested by the Pearson correlation coefficient. The cutoff for significance for all statistical tests was *p* ≤ 0.05.

## 3. Results

The calculated reference intervals for the clinical chemistry analytes independent of sex and age are shown in Table 1A. GGT was only above the cutoff for measurement (<0.1 U/L) in 6 of 39 penguins and was therefore excluded from the results of the present study. The following clinical chemistry analytes were significantly higher in females: GLDH (*p* = 0.0003), ALT (*p* = 0.0005), AP (*p* = 0.0245), AST (*p* = 0.0043), CK (*p* = 0.0016), LDH (*p* < 0.0001), P (*p* = 0.0223), and Na (*p* = 0.0415). For these analytes, the mean, SD, median, minimum, maximum, and the 10% and 90% percentiles were additionally calculated separately for males and females (Table 1B). Calculated reference intervals for haematology are shown in Table 2. There were no significant differences in the haematology analytes between males and females. Cholesterol (*p* = 0.0233; −0.39), WBC (*p* = 0.0208; −0.40), and absolute heterophiles (*p* = 0.0148; −0.42) showed significant negative correlations with the age of the penguins. Separated into the different sexes, females showed a significant negative correlation with age for cholesterol (*p* = 0.0316; −0.51) and absolute heterophiles (*p* = 0.0349; −0.50), while males showed no significant correlations with the age of the individuals. Due to the limited number of samples, no age-specific values were calculated.

## 4. Discussion

Currently, there are limited data available on clinical chemistry and haematology values in Humboldt penguins. Liver-associated enzymes like ALT were higher while AP was lower in the present study in comparison with the values published for wild Humboldt penguins in Chile (Table 3) [1]. Moreno-Salas et al. [4] reported ALT values in captive and wild Humboldt penguins which were slightly higher. AST was also higher in comparison with previously published values for Humboldt penguins [1,2,3,4]. However, AST, ALT, and AP are not only found in the liver but in other organs as well, so it is not clear what led to the different values in the different studies [8]. AP is found in the liver, kidneys, intestines, and bones [8]. ALT is found in almost all tissues. Both enzymes are therefore non-specific indicators in blood testing, and differences in measured values in the present study compared with previous reports can be due to a wide variety of factors like growth, egg laying, and organ disease [6,8]. AST can be increased by both liver and muscle injury. It is therefore possible that the higher AST values measured here were triggered by the handling of the animals for blood collection. This could also explain the distinctly higher CK values measured in the present study compared with previously published values [1,2,4]. The cholesterol levels reported here are in a similar range to the levels published for wild Humboldt penguins [1,2]. The UA values of the present study are also similar to the values which were previously published [4] and slightly higher than the values in Humboldt penguins in Chile [1,3]. Villouta et al. [3] documented decreasing UA values when the animals were kept for a longer period of time, which fits with the distinctly higher UA values reported in another study in wild Humboldt penguins [2]. Reasons for this could be the differences in feed or feed quantity and the time span between the last feed intake and blood collection. A study in African penguins showed a significant postprandial increase in UA, triglycerides, and BA [22]. The published values for Ca and P are different between the two studies in wild Humboldt penguins in Chile and Peru [1,2], but the values of Smith et al. [2] are similar to the present results. The present values for Na and K are near the values published by Wallace et al. [1] and slightly higher than the values by Smith et al. [2]. However, the SD calculated by Smith et al. [2], especially the values for Na and Cl, were quite high, indicating a strong variation between the sampled penguins. Reasons for the differences are likely the different physiological conditions of the penguins; the blood samples analysed by Smith et al. [2] were collected from penguins in November in the second nesting season from adults rearing their chicks, while Wallace et al. [1] collected samples from adult breeding penguins in April and again from non-breeding penguins in September. Moulting could also influence the blood values [23], but the penguins in all studies were outside of the moulting season. The different numbers of collected samples of the various studies could have influenced the results as well. LDH values in our study were comparable to the values reported for captive Humboldt penguins, but distinctly higher than the values measured in wild individuals in Chile [4]. At the time of writing, there were no published values available for amylase, GLDH, CHE, BA, triglycerides, or β-HBS in Humboldt penguins. Amylase is not specific in birds and can increase due to pancreatic diseases, enteritis, and renal diseases [7]. In mammals, GLDH is found in the mitochondria of liver cells and is therefore regarded as a specific indicator for liver cell damage [8]; however, at this time, there are only reference intervals for a few bird species available, and the diagnostic significance of GLDH in birds is therefore not conclusively clarified. Increases in GGT values are considered specific for biliary damage, obstructions, and carcinomas of the bile ducts [8], but a clinical connection in all avian species has not yet been fully clarified. In the present study, GGT was below the cutoff for measurement (<0.1 U/L) in most individuals; therefore, no specific values for reference intervals for Humboldt penguins could be calculated. It is not clear why the GGT was so low in most of the sampled individuals in the present study. Moreno-Salas et al. [4] were able to measure GGT values in Humboldt penguins which differed significantly between penguins in the wild (3.0 ± 2.0 U/L) and those in captivity (13.0 ± 10.0 U/L). CHE has been shown in bearded vultures (*Gypaetus barbatus*) to act as a biomarker for organophosphate and carbamate intoxications [9]. A study in wild Humboldt penguins in Peru indicated that this species is also affected by pesticides, and the authors recommended further monitoring in this species [24]. However, in psittacine birds, CHE blood levels vary depending on species, age, season, and health status of the individual [25]. Phylogenic factors, sex, diet, and stress have also been discussed as influencing factors for CHE in birds [25,26], and detailed studies in most bird species are lacking. BA values of wild adult African penguins during the breeding season were distinctly higher (9.53 ± 16.75 µmol/L) [10] than the values measured in the present study, which could have been influenced by the time span between the last feeding and blood collection [22]—a span which was unknown in the wild penguins and was 18 h for most of the penguins in the present study. The measured β-HBS values of the present study are between the values measured in the two studies of healthy African penguins (0.64 ± 0.25 mmol/L) [27] and (0.94 ± 0.13 mmol/L) [28]. A study of African penguins [28] showed that these values increased in penguins infected with aspergillosis, whereas the penguins in the present study showed no signs of infection in the clinical examination before blood collection.

The PCV and haemoglobin values reported for short-term captive Humboldt penguins [3] and wild and captive Humboldt penguins in Chile [4] are lower in comparison with the values of the present study, but our PCV values are similar to those published for wild individuals in Chile and Peru [1,2], especially in regards to the values reported by Wallace et al. [1]. The WBC was distinctly higher in the present study than most previously published values [2,3,4] but lower than those reported by Wallace et al. [1]. The number of heterophiles was greater in the present study in comparison with previously published values [1,2,3,4], while the number of lymphocytes was smaller compared with the values reported by Wallace et al. [1] and by Moreno-Salas et al. [4] for captive penguins but similar to the values reported by Villouta et al. [3] and by Moreno-Salas et al. [4] for wild penguins. This could be influenced by differences in exposure to pathogens like avian malaria [16] and especially by different levels of acute and chronic stress associated with reproductive cycles, seasonal changes, and injuries, as stress can lead to a shift in the heterophile:lymphocyte ratio [29]. A study in wild Adélie penguins (*Pygoscelis adeliae*) showed that handling at an interval of several days in the breeding season had no significant effect on the corticosterone or heterophile:lymphocyte ratio and that handling and blood collection within less than 5 min had no effect on corticosterone levels [30]. The numbers of monocytes and eosinophiles reported previously have been greater than those found here [1,2,3], although Wallace et al. [1] did not differentiate eosinophiles. Moreno-Salas et al. [4] also found higher monocyte counts but smaller numbers of eosinophiles than in the present study. This could be influenced by parasitic infections, which are more common in wild animals. Basophils are rare in Humboldt penguins [3,4]. Wallace et al. [1] reported the greatest number of basophils of those published, while Smith et al. [2] found no basophils in the blood smears they evaluated, similar to the present study.

A number of sex-specific variations were found for clinical chemistry analytes in the present study but not for haematology. The age of the individual also influenced the quantities of some analytes measured, including the cholesterol levels, the WBC, and absolute heterophiles. Other studies have also found sex- and age-specific variations. A study analysing the variations by year, sex, and age class of beached Magellanic penguins (*Spheniscus magellanicus*) in North Argentina during the non-breeding season found significant variations for AP and CK by year, no sex-specific variations, and significantly higher haematocrit, haemoglobin, erythrocyte, lymphocyte, glucose, UA, total protein, albumin, globulin, and cholesterol values in older juveniles in comparison with younger juveniles, while the younger animals had higher heterophile:lymphocyte ratios and CK values [14]. Another study of free-ranging Magellanic penguins on the Argentinean Patagonian coast found significantly higher PCV, percentage heterophiles and eosinophiles, and cholesterol values in males than in females [11]. In wild adult African penguins, males had significantly higher haematocrit and haemoglobin levels and erythrocyte counts, while females had significantly higher total calcium and inorganic phosphate values [10]. A study in wild nestling Rockhopper penguins (*Eudyptes chrysocome*) found no significant differences between males and females [31]. A study on captive Adélie, Chinstrap (*Pygoscelis antarcticus*), Gentoo (*Pygoscelis papua*), and Macaroni (*Eudyptes chrysolophus*) penguins in Taiwan found significantly lower values for mean corpuscular haemoglobin concentration (MCHC) and urea, higher values for Ca in females, a significant negative correlation with age for erythrocytes, lymphocytes, thrombocytes, AP, CK, LDH, and iron, and a significant positive correlation with age for mean corpuscular volume (MCV), mean corpuscular haemoglobin (MCH), heterophiles, heterophile:lymphocyte ratios, ALT, and chloride [12]. The differences between the studies could have been influenced by the fact that the animals had different ages, were sampled at different locations, at different times, and are of different species and particularly by the fact that most previous studies are based on wild penguins, where many factors like health status and food intake are not exactly known.

## 5. Conclusions

The present study provides the first reference intervals for a wide range of clinical chemistry and haematology analytes of a large zoological collection of Humboldt penguins. Some differences could be found between males and females for clinical chemistry analytes but not for haematology. The age of the individuals also influences some analytes and should be considered when interpreting laboratory results. These results are a basis for further studies and will help to improve the medical management of captive as well as wild penguins.

## Figures and Tables

**Table 1 animals-13-03570-t001:** Clinical chemistry analytes of a single captive collection of Humboldt penguins (*Spheniscus humboldti*) in Germany (*n* = 39). For all analytes, the mean, standard deviation (SD), median, minimum, maximum, and the 10% and 90% percentiles, as well as the lower and upper reference intervals, were calculated for all animals (All) (**A**). For those analytes in which significant differences between males (*n* = 19) and females (*n* = 20) were detected, the specific mean, SD, median, minimum, maximum, and the 10% and 90% percentiles were additionally calculated for each sex separately (**B**).

A																				
Analyte	Unit		Sex	Mean		SD		Median		Minimum		Maximum		10% Percentile	90% Percentile		Lower Reference Interval Limit		Upper Reference Interval Limit	
Amylase	U/L		All	1370.18 ^a^		336.05		1386.0		747.0		2286.0		939.0	1821.0		812.0 (732.8–913.3)		2185.4 (1952.1–2420.8)	
Glutamate dehydrogenase	U/L		All	10.23 ^b^		9.24		7.50		0.05		39.3		1.10	24.9		0.2 (0.0–0.8)		36.8 (27.6–46.7)	
Alanine transaminase	U/L		All	121.32 ^b^		79.35		94.8		46.0		438.3		57.4	205.5		46.5 (43.1–52.5)		338.4 (252.0–442.5)	
Alkaline phosphatase	U/L		All	87.44 ^b^		28.79		81.0		44.0		183.0		54.0	141.0		46.6 (41.9–52.8)		165.3 (139.0–195.1)	
Cholinesterase	U/L		All	4598.92 ^b^		1090.42		4374.0		3312.0		8310.0		3429.0	6414.0		3071.7 (2876.5–3321.9)		7381.6 (6409.1–8562.5)	
Bile acids	µmol/L		All	1.06 ^b^		0.92		0.99		0.05		3.93		0.05	2.23		0.05 (0.0–0.1)		4.2 (3.4–5.0)	
Aspartate aminotransferase	U/L		All	365.75 ^b^		247.24		275.2		153.6		1295.4		194.0	631.8		158.4 (149.0–173.5)		1085.0 (754.1–1551.2)	
Creatine kinase	U/L		All	4381.51 ^b^		6240.13		1900.0		244.0		26,652.0		529.0	11,259.0		254.5 (242.8–317.5)		21,908.2 (14,087.8–31,693.5)	
Lactate dehydrogenase	U/L		All	751.03 ^b^		563.59		588.2		173.1		2555.7		302.6	1513.5		212.5 (190.2–250.8)		2707.6 (1852.9–3843.5)	
Cholesterol	mmol/L		All	5.84 ^a^		0.95		5.80		3.90		8.10		4.80	7.40		4.1 (3.7–4.4)		7.9 (7.4–8.5)	
Triglyceride	mmol/L		All	0.78 ^b^		0.30		0.67		0.47		1.62		0.49	1.33		0.5 (0.4–0.6)		1.5 (1.3–1.5)	
β-hydroxybutyric acid	mmol/L		All	0.85 ^b^		0.47		0.77		0.24		2.79		0.39	1.44		0.3 (0.2–0.3)		2.1 (1.7–2.5)	
Uric acid	µmol/L		All	428.87 ^b^		184.77		397.0		72.0		920.0		226.0	691.0		125.0 (96.7–169.9)		909.6 (773.5–1043.5)	
Calcium	mmol/L		All	2.70 ^b^		0.33		2.60		2.30		3.60		2.40	3.30		2.3 (2.3–2.4)		3.8 (3.3–4.7)	
Inorganic Phosphorus	mmol/L		All	1.17 ^b^		0.56		1.00		0.50		2.70		0.60	2.10		0.4 (0.4–0.5)		2.6 (2.1–3.0)	
Sodium	mmol/L		All	158.34 ^b^		4.76		157.0		152.0		175.0		154.0	164.0		148.5 (146.5–151.3)		168.2 (163.4–172.5)	
Potassium	mmol/L		All	3.82 ^b^		0.90		3.60		2.50		7.20		2.90	4.80		2.6 (2.5–2.8)		6.3 (5.2–8.0)	
Chloride	mmol/L		All	113.16 ^b^		2.83		112.5		109.0		119.0		110.0	117.0		107.3 (106.6–108.4)		119.0 (117.5–120.4)	
**B**																				
**Analyte**		**Unit**			**Sex**		**Mean**		**SD**		**Median**		**Minimum**			**Maximum**		**10% Percentile**		**90% Percentile**
Glutamate dehydrogenase		U/L			male		5.43 ^b^		4.27		4.7		0.05			14.9		0.05		14.6
					female		14.79 ^b^		10.43		11.3		2.20			39.3		5.30		32.55
Alanine transaminase		U/L			male		82.59 ^b^		28.43		80.0		46.0			175.4		48.2		109.1
					female		158.11 ^b^		94.35		135.9		57.4			438.3		71.05		291.0
Alkaline phosphatase		U/L			male		75.00 ^a^		20.56		78.0		44.0			113.0		45.0		113.0
					female		99.25 ^b^		30.93		83.5		73.0			183.0		74.5		145.0
Aspartate aminotransferase		U/L			male		263.73 ^b^		104.55		238.4		153.6			625.0		164.4		375.2
					female		462.67 ^b^		302.66		345.5		198.0			1295.4		231.6		979.95
Creatine kinase		U/L			male		1909.74 ^b^		2206.87		1071.0		244.0			8919.0		282.0		6075.0
					female		6729.70 ^b^		7832.66		3082.5		641.0			26,652.0		1123.5		21,684.0
Lactate dehydrogenase		U/L			male		457.11 ^b^		235.51		386.6		173.1			1199.7		215.6		791.1
					female		1030.26 ^b^		643.63		719.25		425.2			25,555.7		525.85		2273.4
Inorganic Phosphorus		mmol/L			male		0.94 ^b^		0.34		0.90		0.50			1.70		0.60		1.70
					female		1.39^a^		0.65		1.25		0.50			2.70		0.65		2.40
Sodium		mmol/L			male		156.74 ^a^		2.96		157.0		152.0			164.0		153.0		160.0
					female		160.25 ^b^		5.80		158.5		154.0			175.0		155.0		170.0

^a^ normally distributed; ^b^ not normally distributed. Due to the limited number of samples, no sex-specific reference intervals could be calculated.

**Table 2 animals-13-03570-t002:** Haematological analytes of a single captive collection of Humboldt penguins (*Spheniscus humboldti*) in Germany (*n* = 39). For all analytes the mean, standard deviation (SD), median, minimum, maximum, and the 10% and 90% percentiles, as well as the lower and upper reference interval limits, were calculated for all animals. Since there were no significant differences between males and females, no sex-specific values were calculated.

Analyte	Unit	Mean	SD	Median	Minimum	Maximum	10% Percentile	90% Percentile	Lower Reference Interval Limit	Upper Reference Interval Limit
Packed cell volume	L/L	0.50 ^a^	0.06	0.50	0.36	0.64	0.40	0.59	0.4 (0.3–0.4)	0.6 (0.6–0.7)
Haemoglobin	g/L	154.24 ^a^	14.94	154.0	125.0	189.0	131.0	175.0	123.4 (115.9–130.7)	184.8 (177.7–192.1)
White blood cell count	G/L	19.14 ^b^	5.25	17.8	12.4	39.8	13.2	24.2	12.2 (11.5–13.2)	32.8 (28.2–38.0)
Heterophiles	%	73.41 ^b^	7.80	74.00	57.0	87.0	59.0	83.0	55.7 (50.3–60.4)	87.9 (85.0–90.7)
	G/L	14.20 ^b^	4.92	13.69	7.79	34.63	8.66	18.72	7.6 (6.7–8.7)	26.4 (22.1–31.8)
Lymphocytes	%	23.26 ^b^	7.32	21.0	10.0	40.0	13.0	35.0	10.2 (8.0–12.6)	39.9 (35.6–44.4)
	G/L	4.30 ^b^	1.37	4.10	1.83	8.35	2.58	6.24	2.1 (1.8–2.5)	7.8 (6.7–8.8)
Monocytes	%	0.49 ^b^	0.76	0.0	0.0	3.0	0.0	2.0	0.0 (0.0–0.1)	3.0 (2.6–3.4)
	G/L	0.09 ^b^	0.15	0.0	0.0	0.53	0.0	0.34	0.0 (0.0–0.1)	0.51 (0.48–0.54)
Eosinophiles	%	2.85 ^b^	3.07	1.0	0.0	14.0	0.0	7.0	0.0 (0.0–0.1)	12.0 (11.0–13.0)
	G/L	0.54 ^b^	0.60	0.24	0.0	3.08	0.0	1.16	0.0 (0.0–0.1)	1.8 (1.3–2.4)
Basophiles	%	0.0	0.0	0.0	0.0	0.0	0.0	0.0	0.0	0.0
	G/L	0.0	0.0	0.0	0.0	0.0	0.0	0.0	0.0	0.0

^a^ normally distributed; ^b^ not normally distributed.

**Table 3 animals-13-03570-t003:** Mean and standard deviations (SD) for various blood analytes calculated in the present study in comparison with previously published values for Humboldt penguins (*Spheniscus humboldti*).

	Unit	Present Study	Wild Humboldt Penguins in Chile [1]	Humboldt Penguins 15 Weeks after Capture in the Wild in Chile [3]	Wild Humboldt Penguins in Peru [2]	Wild Humboldt Penguins in Chile [4]	Captive Humboldt Penguins from Different Zoos in Chile [4]
		*n* = 39	*n* = 51	*n* = 14	*n* = 90	*n* = 21	*n* = 21
		Mean ± SD	Mean ± SD	Mean ± SD	Mean ± SD	Mean ± SD	Mean ± SD
ALT	U/L	121.32 ± 79.35	58.80 ± 37.66	n.a	n.a	137.0 ± 33.0	138.0 ± 32.0
AP	U/L	87.44 ± 28.79	101.2 ± 56.19	n.a	n.a	37.0 ± 15.0	113.0 ± 82.0
AST	U/L	365.75 ± 247.24	206.8 ± 88.23	200.9 ± 67.0	208.1 ± 118.8	178.0 ± 92.0	145.0 ± 76.0
CK	U/L	4381.51 ± 6240.13	222.4 ± 148.0	n.a	361.5 ± 172.9	47.0 ± 31.0	59.0 ± 79.0
LDH	U/L	751.03 ± 563.59	n.a	n.a	n.a	285.0 ± 172.0	732.0 ± 434.0
Cholesterol	mmol/L	5.84 ± 0.95	5.49 ± 1.37	n.a	5.02 ± 0.91	n.a	n.a
Uric Acid	µmol/L	428.87 ± 184.77	377.0 ± 184.0	374.0 ± 84.0	810.0 ± 610.0	430.0 ± 120.0	440.0 ± 210.0
Calcium	mmol/L	2.70 ± 0.33	3.52 ± 1.30	n.a	2.50 ± 0.34	n.a	n.a
Inorganic Phosphorus	mmol/L	1.17 ± 0.56	1.37 ± 0.68	n.a	1.19 ± 0.40	n.a	n.a
Sodium	mmol/L	158.34 ± 4.76	154.0 ± 3.48	n.a	149.6 ± 13.1	n.a	n.a
Potassium	mmol/L	3.82 ± 0.90	3.47 ± 0.72	n.a	2.82 ± 0.70	n.a	n.a
Chloride	mmol/L	113.16 ± 2.83	n.a	n.a	108.0 ± 10.4	n.a	n.a
PCV	L/L	0.50 ± 0.06	0.49 ± 0.05	0.43 ± 0.04	0.47 ± 0.04	0.43 ± 0.09	0.46 ± 0.09
Haemoglobin	g/L	154.24 ± 14.94	n.a	150.0 ± 16.0	n.a	n.a	n.a
WBC	G/L	19.14 ± 5.25	23.25 ± 7.08	13.0 ± 3.7	12.86 ± 11.5	8.36 ± 1.58	12.50 ± 4.20
Heterophiles	%	73.41 ± 7.80	n.a	62.8 ± 9.7	45.6 ± 10.6	n.a	n.a
	G/L	14.20 ± 4.92	12.20 ± 3.83	8.0 ± 3.2	n.a	5.6 ± 1.2	6.7 ± 3.5
Lymphocytes	%	23.26 ± 7.32	n.a	22.8 ± 9.5	47.3 ± 10.0	n.a	n.a
	G/L	4.30 ± 1.37	8.06 ± 3.73	2.8 ± 1.4	n.a	1.8 ± 0.8	4.7 ± 2.2
Monocytes	%	0.49 ± 0.76	n.a	4.4 ± 4.1	3.2 ± 2.4	n.a	n.a
	G/L	0.09 ± 0.15	0.76 ± 0.79	0.6 ± 0.8	n.a	0.32 ± 0.29	0.68 ± 0.73
Eosinophiles	%	2.85 ± 3.07	n.a	10.3 ± 8.6	3.6 ± 4.0	n.a	n.a
	G/L	0.54 ± 0.60	n.a	1.1 ± 0.9	n.a	0.47 ± 0.26	0.28 ± 0.22
Basophiles	%	0.0 ± 0.0	n.a	0.1 ± 0.3	0.0	n.a	n.a
	G/L	0.0 ± 0.0	0.40 ± 0.31	0.01 ± 0.05	n.a	0.005 ± 0.016	0.01 ± 0.03

n.a: not analysed.

## Data Availability

The data that support the findings of this study are available from the corresponding author upon reasonable request.

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
