# Peer review of "Clinical Chemistry and Haematology Values of a Captive Population of Humboldt Penguins (Spheniscus humboldti)"

_animals, 2023, doi:10.3390/ani13223570_

Round 1

Reviewer 1 Report

Comments and Suggestions for Authors

This article describes the hematology and clinical chemistry in a single population of Humboldt penguins in a zoologic facility and compares the obtained values between males and females and with those reported in wild Humboldt penguins.  As current information regarding clinical reference ranges for this species is limited, this article would be of interest to those involved in the care of avian species in a zoologic setting as well as those involved in zoologic and wild avian medicine and research.

Overall, the authors do a good job of addressing the problem of limited availability of reference ranges for this species in a zoologic setting, as well as delving into the possible reasons for the observed differences between male and female penguins and between this captive population and previously reported values for wild populations.  The introduction clearly and concisely states the deficiency in information this study intends to address and the potential benefit to the study and care of captive and wild penguins.  The methods are sufficiently detailed and well organized.  The results are also well organized, Table 3 nicely summarizes the current knowledge regarding clinical values in this species.  The discussion is thorough and addresses the challenges of comparing the values of wild and captive populations.  I have a few comments pertaining to minor clarifications and organizational points. In the Methods section in Line 56 where the age range of the animals is stated, please clarify if was there no/any significant difference in ages between males and females. The results are well organized, although Table 1 is a bit overwhelming due to its size.  Perhaps if there is space it could be broken up by having the analytes that were significantly different in males vs females in a separate table.  Might also note if there was no/any significant age difference between males and females in the results paragraph.

Author Response

This article describes the hematology and clinical chemistry in a single population of Humboldt penguins in a zoologic facility and compares the obtained values between males and females and with those reported in wild Humboldt penguins.  As current information regarding clinical reference ranges for this species is limited, this article would be of interest to those involved in the care of avian species in a zoologic setting as well as those involved in zoologic and wild avian medicine and research.

Overall, the authors do a good job of addressing the problem of limited availability of reference ranges for this species in a zoologic setting, as well as delving into the possible reasons for the observed differences between male and female penguins and between this captive population and previously reported values for wild populations.  The introduction clearly and concisely states the deficiency in information this study intends to address and the potential benefit to the study and care of captive and wild penguins.  The methods are sufficiently detailed and well organized.  The results are also well organized, Table 3 nicely summarizes the current knowledge regarding clinical values in this species.  The discussion is thorough and addresses the challenges of comparing the values of wild and captive populations. 

Thank you very much for the words of praise and the great feedback to our study.

I have a few comments pertaining to minor clarifications and organizational points. In the Methods section in Line 56 where the age range of the animals is stated, please clarify if was there no/any significant difference in ages between males and females.

When we looked at the age correlation separated by sex, we found a significant correlation for cholesterol and absolute heterophiles in females and no significant age corelations in males. This has now been added to the result section.

The results are well organized, although Table 1 is a bit overwhelming due to its size.  Perhaps if there is space it could be broken up by having the analytes that were significantly different in males vs females in a separate table.  Might also note if there was no/any significant age difference between males and females in the results paragraph.

Table 1 Has been split into 1A showing calculations for all penguins, independent of sex and 1B showing calculations for individual sexes as suggested.

Reviewer 2 Report

Comments and Suggestions for Authors

General comments.

This study provides reference data for Humboldt penguins in captivity at the zoo. Biochemical and hematological studies of captive animals are important because they provide information from individuals under controlled conditions that are useful for future comparison and interpretation with data from wild animals. The paper is well-written, and the overall approach is understandable. The techniques used are common in hematology and biochemistry.

The aim of this paper is to establish the reference values of apparently healthy Humboldt penguin individuals in captivity at the zoo. While it is hypothesized that there are differences between males and females and between the ages of individuals, it is not clear what this hypothesis is based on, and no predictions are made.

For baseline reports, it is desirable to have a larger sample than reported, if possible, to obtain larger intervals, especially if sex and age are to be taken into account. Here, some differences were found in biochemical parameters between the sexes, but not in haematological parameters. Some differences were found between individuals of different ages. Perhaps these differences could be more robust with a higher N, or conversely, not be differences.

Perhaps a paragraph could be added in the introduction explaining in general terms why these analytes were chosen for this study and what information they provide.

A weakness of this work is that penguin handling time data were not recorded or reported, which is a common cause of variation in the physiological parameters of many animals, especially those related to stress.

I think that the authors should consult more literature for comparison with other penguin species in captivity and/or under controlled conditions. There is also a lot of literature on other penguin species.

I suggest that when a reference is cited, it is made clear what is being studied in that reference so that it is easier to interpret the comparison. For example, on pages 166-169, what species, where, and under what circumstances did the cited authors work with?

I suggest that all information is given in the tables. Since this is a paper in which reference values are given, it is ideal to indicate all the analytes for males and females and the different age classes, even if the differences were not significant, and to identify those where the differences are significant (to be interpreted later).

Minor comments:   14-16: It is striking that the study is limited to zoos, precisely where data collection is more feasible. I would rather think of the opposite, that it is difficult to obtain reference values in wild animals.   71- Clarify when the breeding and molting seasons   117- Is there a specific interpretation as to why the GGT value is higher in 6 animals?   152-158- The authors suggest that the variation in enzyme levels may be due to the handling of the penguins during blood collection. A highly recommended option is to record the handling time of the individuals and thus see the variation in values in relation to the times measured. This would also provide timing options for the handling of the animals and the sample collection process to obtain robust, reliable, and comparable values. In addition, it is a practice that may be easier to carry out in captive animals.   180-183: Some suggestions for the interpretation of the parameters? Even if based on values for other species. If not, it is not clear why they chose to give values that are not diagnostic for birds. Perhaps this should be explained in the introduction; the analytes chosen and why.   190: Change the word animals to individuals.   191-192: An explanation of the different values that might provoke discussion.   205-207: The individuals sampled were considered healthy, but was a pathogenic test done to support this? I might assume that it is easier to monitor the health of penguins in captivity than penguins in the wild.   216. Change penguines for penguins. Review the entire manuscript for these kinds of errors.   220-222: This could easily have been included in this baseline study.   244-248. This is something that can only be checked for individual animals in the zoo. Perhaps some information about the type of food, hours, etc. could be included for informational purposes.   Bibliography: See that some reference numbers do not match the cited bibliography; cases 1 and 2. Check the references.

Author Response

This study provides reference data for Humboldt penguins in captivity at the zoo. Biochemical and hematological studies of captive animals are important because they provide information from individuals under controlled conditions that are useful for future comparison and interpretation with data from wild animals. The paper is well-written, and the overall approach is understandable. The techniques used are common in hematology and biochemistry.

The aim of this paper is to establish the reference values of apparently healthy Humboldt penguin individuals in captivity at the zoo. While it is hypothesized that there are differences between males and females and between the ages of individuals, it is not clear what this hypothesis is based on, and no predictions are made.

An additional paragraph has been added in the introduction to help clarify the basis for the hypotheses.

For baseline reports, it is desirable to have a larger sample than reported, if possible, to obtain larger intervals, especially if sex and age are to be taken into account. Here, some differences were found in biochemical parameters between the sexes, but not in haematological parameters. Some differences were found between individuals of different ages. Perhaps these differences could be more robust with a higher N, or conversely, not be differences.

Yes, this is correct, but the problem is that the values could also be different between different zoological collections or populations. Our study is based on animals from a single zoological collection with constant husbandry and feeding conditions. We collected samples form the whole population. As a next step, it would be interesting to compare these results with those from other populations in order to optimize available ranges and aid in interpretation. However, that goes beyond the scope of the present study. This limitation and the need for further data is discussed in the paper.

Perhaps a paragraph could be added in the introduction explaining in general terms why these analytes were chosen for this study and what information they provide.

This has been added.

A weakness of this work is that penguin handling time data were not recorded or reported, which is a common cause of variation in the physiological parameters of many animals, especially those related to stress.

The handling time has been added in the material and method section.

I think that the authors should consult more literature for comparison with other penguin species in captivity and/or under controlled conditions. There is also a lot of literature on other penguin species.

Additional literature has been added.

I suggest that when a reference is cited, it is made clear what is being studied in that reference so that it is easier to interpret the comparison. For example, on pages 166-169, what species, where, and under what circumstances did the cited authors work with?

The cited studies are based on wild Humbold penguins in Chile and Peru. This has been clarified in the text.

I suggest that all information is given in the tables. Since this is a paper in which reference values are given, it is ideal to indicate all the analytes for males and females and the different age classes, even if the differences were not significant, and to identify those where the differences are significant (to be interpreted later).

This would make the table much larger and more confusing, which is contrary to the comment of reviewer 1. The editor should decide if we should divide the data up more than has currently been done. 

Minor comments:   

14-16: It is striking that the study is limited to zoos, precisely where data collection is more feasible. I would rather think of the opposite, that it is difficult to obtain reference values in wild animals.   

This sentence refers to the published literature, in which more data on wild pinguins is available. But you are correct that it is generally easier to collect samples from captive populations. The sentence has been rewritten accordingly.

71- Clarify when the breeding and molting seasons   

Have been added in the materials and methods section.

117- Is there a specific interpretation as to why the GGT value is higher in 6 animals? 

Unfortunately, no, all were clinically healthy and showed no other changes until now.  This has been added in the discussion.

152-58 The authors suggest that the variation in enzyme levels may be due to the handling of the penguins during blood collection. A highly recommended option is to record the handling time of the individuals and thus see the variation in values in relation to the times measured. This would also provide timing options for the handling of the animals and the sample collection process to obtain robust, reliable, and comparable values. In addition, it is a practicethat may be easier to carry out in captive animals.   

Yes, this is correct, we have added the handling time for the present study in the material and method section now.

180-83: Some suggestions for the interpretation of the parameters? Even if based on values for other species. If not, it is not clear why they chose to give values that are not diagnostic for birds. Perhaps this should be explained in the introduction; the analytes chosen and why.   

A paragraph has been added in the introduction as suggested.

190: Change the word animals to individuals.  

Has been corrected.

191-192: An explanation of the different values that might provoke discussion.  

Yes, but there are to little data at this time to that would help with this explanation.

205-07: The individuals sampled were considered healthy, but was a pathogenic test done to support this? I might assume that it is easier to monitor the health of penguins in captivity than penguins in the wild.  

Endo- and Ektoparasittes were checked, this has been added in the materials and methods section.

  1. Changepenguines for penguins. Review the entire manuscript for these kinds of errors. 

This mistake has been corrected.  

 220-22: This could easily have been included in this baseline study.   

Unfortunately, we did not measure other stress indicators such as cortisol and tried to sample all animals as quickly and evenly as possible.

244-48. This is something that can only be checked for individual animals in the zoo. Perhaps some information about the type of food, hours, etc. could be included for informational purposes.  

Yes, we agree on this point, but most of the cited studies are from wild individuals, so this information is not provided. In the study by Moreno-Salas, no specific information was provided.

Bibliography: See that some reference numbers do not match the cited bibliography; cases 1 and 2. Check the references.

Have been checked and corrected.

Reviewer 3 Report

Comments and Suggestions for Authors

This manuscript is well-written and is of clinical relevance. Hematological and biochemical reference intervals are essential for clinicians working with this species. A weakness in an otherwise interesting manuscript is the lack of values for total protein and protein fractions.

Author Response

This manuscript presents new reference intervals for some known haematological and biochemical parameters for Humboldt penguins but also, for new parameters with clinical importance, The study has been well designed and the results are presented in clear way. Information about total protein, albumin and globulin fractions is missing. This is a weakness of this study as these parameters are of clinical relevance and often looked for.

Thank you very much for your great feedback. We agree that the total protein and the protein fractions are very important. Due to the fact that the albumin measurement by BCG, which is the standard test used by most autoanalyzer’s, is not valid for avians and reptiles (Cray 2023 Re-examination of BCG albumin: Reference intervals for this method may not be valid as reported in other avian and reptilian species), we decided to measure these values by capillary zone electrophoresis and the data from those analyses have been written up in a separate manuscript which is also under review at this time.

The discussion addresses all the questions raised by the comparison with previous studies and is supported by the literature review.

Thank you very much for you comment.

Some direct comments are listed below.

Line 81-82: Did you refrigerated/freeze the plasma? And the whole blood tube? Have you sent the tubes at room temperature to the lab or refrigerated?

The samples were sent refrigerated to the lab, this information has been added to the text.

Line 97-98: please explain how you determined the total WBC count in the text.

The calculation has been added.

CK: Please review your data, the SD is bigger than the mean and median this will mean the probability of negative values for this parameter.

Eosinophiles: Please review your data, the SD is bigger than the mean and median this will mean the probability of negative values for this parameter.

We have checked the data again. For these two analytes, the measured values fluctuate very strongly and have such a large variance that the coefficient of variance is greater than 1 and the mean value is smaller than the SD. Additionally the data for these two analytes are not normally distributed.

Reviewer 4 Report

Comments and Suggestions for Authors

In this manuscript, the authors have extensively studied specific reference intervals for a series of clinical chemistry and hematology analytes for captive Humboldt penguins. 

This study establishes the initial reference ranges for a comprehensive set of clinical chemistry and hematology measurements from a substantial group of Humboldt penguins within a zoo setting. While there were variations noted between males and females in clinical chemistry parameters, no such distinctions were observed in hematology. Additionally, the age of the penguins was found to impact certain measurements, highlighting the importance of age when interpreting lab findings. I consider that these findings serve as a fundamental framework for future investigations and will enhance the healthcare of both captive and wild penguin populations.

Although there have been similar previous studies, none of them included as many analytical parameters. The authors have conducted a comprehensive comparison of the results obtained here with those from previous studies.

This manuscript is well structured and synthesized; however, some issues need to be addressed to ensure that the correct ideas are conveyed:

1. Material and Methods: 

Consideration of healthy animals: Were infectious diseases evaluated?

What has been done with the outliers? If healthy, the argument is to keep them, but you have to state it

2. Results: 

Could you please add the results of data distribution?

Table 1 seems difficult to understand to me.

3. Discussion: 

line 154: please, name the most important factors in reference 10 and explain how those factors may be influencing your results

line 157: Ck values were extremely high compared to previous studies. Were the animals from your study used to manual restraint? Normally, captive penguins are quite accustomed to manual restraint in a zoological institution. According to reference 10: 'When there is a concurrent elevation in CK activity, increases in AST and LDH are more likely caused by muscle damage.' Therefore, it appears that your animals experienced considerable stress during sampling.  If so, we would expect more changes in the analytes assessed. But on the contrary, if you assume that handling was not particularly challenging, how would you explain these results? 

Lines 176-177 : You cannot use molting as a justification for this variation in the results if, as clearly indicated below, the animals were not molting.

Lines 206-207: Once again, how were infectious diseases ruled out in the study animals? Was it solely based on clinical exams?

Lines 217-220: It's important to note that the response to stress can vary from animal to animal, and not all individuals will experience the same changes in these parameters. Furthermore, chronic and prolonged stress can have more significant effects on health compared to acute and temporary stress. In addition, stress could have effects on other analytical parameters, not just on heterophils. Then, I am really worried about having tested stressed animals. On the other hand, if the age seemed to influence the heterophiles count, this might also be a justification to your increased levels to that of other authors

Author Response

In this manuscript, the authors have extensively studied specific reference intervals for a series of clinical chemistry and hematology analytes for captive Humboldt penguins.

This study establishes the initial reference ranges for a comprehensive set of clinical chemistry and hematology measurements from a substantial group of Humboldt penguins within a zoo setting. While there were variations noted between males and females in clinical chemistry parameters, no such distinctions were observed in hematology. Additionally, the age of the penguins was found to impact certain measurements, highlighting the importance of age when interpreting lab findings. I consider that these findings serve as a fundamental framework for future investigations and will enhance the healthcare of both captive and wild penguin populations.

Although there have been similar previous studies, none of them included as many analytical parameters. The authors have conducted a comprehensive comparison of the results obtained here with those from previous studies.

Thank you very much for your great feedback.

This manuscript is well structured and synthesized; however, some issues need to be addressed to ensure that the correct ideas are conveyed:

  1. Material and Methods:

Consideration of healthy animals: Were infectious diseases evaluated?

Have been added in the materials and methods section.

What has been done with the outliers? If healthy, the argument is to keep them, but you have to state it

An additional sentence has been added in the M+M section.

  1. Results:

Could you please add the results of data distribution?

The results are included in Table 1 as letters after the mean value.

Table 1 seems difficult to understand to me.

The table was revised and divided into two tables.

  1. Discussion:

Line 154: please, name the most important factors in reference 10 and explain how those factors may be influencing your results

Has been added.

Line 157: Ck values were extremely high compared to previous studies. Were the animals from your study used to manual restraint? Normally, captive penguins are quite accustomed to manual restraint in a zoological institution. According to reference 10: 'When there is a concurrent elevation in CK activity, increases in AST and LDH are more likely caused by muscle damage.' Therefore, it appears that your animals experienced considerable stress during sampling.  If so, we would expect more changes in the analytes assessed. But on the contrary, if you assume that handling was not particularly challenging, how would you explain these results?

We have tried to handle the animals as short as possible under 5 min to avoid stress, but we assume that the manual fixation has already led to the higher values. A local tissue irritation at the blood collection site may also have increased the values.

Lines 176-177 : You cannot use molting as a justification for this variation in the results if, as clearly indicated below, the animals were not molting.

This is true, and should only be listed as a possible influencing factor, as the animals were therefore sampled specifically outside of this time.

Lines 206-207: Once again, how were infectious diseases ruled out in the study animals? Was it solely based on clinical exams?

The diagnosis of aspergillosis is very difficult on living bird, the animals were not x-rayed or endoscoped, the exclusion is based on the clinic before, during and after blood sampling. In addition, electrophoresis was performed, which was also unremarkable.

Lines 217-220: It's important to note that the response to stress can vary from animal to animal, and not all individuals will experience the same changes in these parameters. Furthermore, chronic and prolonged stress can have more significant effects on health compared to acute and temporary stress. In addition, stress could have effects on other analytical parameters, not just on heterophils. Then, I am really worried about having tested stressed animals. On the other hand, if the age seemed to influence the heterophiles count, this might also be a justification to your increased levels to that of other authors.

We agree with this and have added an additional sentence, the influences of age are also discussed in line 274 to 296.

Round 2

Reviewer 2 Report

Comments and Suggestions for Authors

The authors have incorporated most of the suggestions made. The hypothesis they are proposing and their predictions are still not clear to me, especially because they are based on differences in the sexes, which they then dismiss due to the sample size. But in general, the manuscript now flows better.

Author Response

We have read through the manuscript in detail, but unfortunately cannot find a section where we present this contradiction. In the results section as well as in the discussion we refer to the differences found between the sexes and also write in the conclusion that there are sex-dependent differences for the clinical chemistry analytes. The fact that there were insufficient samples to establish full reference intervals based on sex does not mean that differentiating between the sexes and documenting differences is not worthwhile.

Reviewer 4 Report

Comments and Suggestions for Authors

All the comments have been correctly adressed. Thank you very much for your new version of the manuscript.

Author Response

Thank you for taking the time to review our manuscript and helping to improve it with your comments.